

# Regional Ionosphere Mapping Using Zero Difference GPS Carrier Phase

Heba Tawfeek[1], Ahmed Sedeek[2], Mostafa Rabah[3], Gamal El-Fiky[1]

1. Faculty of Engineering, Zagazig University, Egypt,
2. Higher Institute of Engineering and Technology, El-Behira, Egypt,
3. Benha Faculty of Engineering, Benha University, Egypt.

## Abstract

Ionospheric delay, can be derived from dual frequency Global Navigation Satellite Systems (GNSS) signals, and then converted into the Vertical Total Electron Contents (VTEC) along the signal path. Various models were devised to calculate VTEC, such as spherical harmonics model and Taylor Series Expansion model. According to density of electron ionosphere can be divided into several layers where electrons reaches peak value at altitude of 350 km. for two-dimension ionosphere modeling, the ionosphere is assumed to be concentrated on a spherical shell of a thin layer located at the altitude of about 350 km above Earth.

The primary goal of the current research to develop an algorithm capable of producing VTEC maps on an hourly basis, using carrier phase observations from dual frequency GPS receiver. The developed algorithm uses a single GPS station (Zero-difference) to map VTEC over a regional area.

The carrier phase measurements are much more precise than the code pseudorange measurements, but they contain an ambiguous term. If such ambiguities are fixed, thence the carrier phase measurements become as unambiguous pseudoranges, but accurate at the level of few millimeters.

In current research Sequential Least Square Adjustment (SLSA) was considered to fix ambiguity term in carrier phase observations. The proposed algorithm was written using MATLAB and Called (ZDPID). Two GPS stations (ANKR and BSHM) were used from IGS network to evaluate the developed code, VTEC values were estimated over these two stations. Results of the proposed algorithm were compared with the Global Ionosphere Maps (GIMs), which is generally used as a reference. The results show that the mean difference between VTEC from GIM and estimated VTEC at ANKR station is ranging from -2.1 to 3.67 TECU and its RMS is 0.44. The mean difference between VTEC from GIM and estimated VTEC at BSHM station is ranging from -0.29 to 3.65 TECU and its RMS is 0.38. Another three GPS stations in Egypt were used to generate regional ionosphere maps over Nile Delta, Egypt. The mean differences between VTEC from GIM and estimated VTEC at SAID station is ranging from - 1.1 to 3.69 TECU and its RMS is 0.37, from -1.29 to 3.27 TECU for HELW station with RMS equal 0.39, and from 0.2 to 4.2 TECU for BORG station with RMS equal 0.46. Therefore, the proposed algorithm can be used to estimate VTEC efficiently.



Keywords: GIMs, Ionosphere mapping, PPP, VTEC.

## 1. Introduction

due to the global system coverage and multiple frequency data available via a world-wide network of GNSS stations
GNSS where considered as a reach source of information about ionosphere of the earth. Since the ionosphere is a dispersive
medium, this multiple frequency, TEC data can be derived along the line-of sight between a given satellite and receiver. By
estimating *VTEC* values from a dual frequency GPS receiver. Dual-frequency GPS receiver demonstrate the number of
electrons in the ionosphere layer in a column of 1 m$^2$ cross-section, which is called the Slant Total Electron Content (*STEC*)
which is extending along the ray-path of the signal between the satellite and the receiver.
different models were used to estimate the ionospheric delay and *STEC*. These models based on spherical harmonic
expansions in the global or regional station scale (Schaer, 1999, and Wielgosz et al., 2003a). Local models based on two-
dimensional Taylor series expansions (e.g. Komjathy, 1997, Jakobsen et al. 2010, Deng et al 2009, and Masaharu et al. 2013).
Ionospheric mapping is defined as a technique applying simultaneously measured TEC values to generate VTEC maps
referred to a specific time epoch (Stanislawska et. al. 2000). Several studies have been performed for regional ionosphere
mapping (Wielgosz et. al. 2003b; Nohutcu et. al. 2010; Salih Alcay et al. 2012). There are several groups which are capable
of producing regional and/or global ionospheric maps like IGS Processing Center at the Astronomical Institute of the
University of Bern, the Orbit Attitude Division of the European Space Operations Centre, and At the GPS Network and
Operations Group of the Jet Propulsion Laboratory, the stations from the IGS network are used to produce regional and global
ionospheric maps (Feltens et al., 1996). global ionospheric maps are routinely produced and made available on the Internet
(Sardon et al., 1995; Jakowski et al., 1996).
Due to the lack of GPS stations over the equatorial, North Africa and Atlantic in IGS network, a proposed algorithm
is suggested to be able to produce ionospheric maps over any other GPS station accurately.
The proposed algorithm in this study is based mainly upon utilizing dual frequency GPS carrier phase observation of a single
station (Zero Differenced) to produce Ionospheric maps under MATLAB environment. To overcome the ambiguous phase
observations, a Sequential Least Square Adjustment (*SLSA)* is considered to fix the ambiguity term.
To evaluate the proposed algorithm two stations from IGS network were considered, and the resulted ionosphere
maps were compared with the Global Ionosphere Maps (GIMs), which is generally used as a reference. Based upon the test
results, the Ionospheric activity over Nile Delta, Egypt, was investigated using three GPS stations. The used GPS data is a
part of the regional GPS network operated by National Research Institute of Astronomy and Geophysics (NRIAG).

## 2. GPS Observations

dual-frequency GPS observations consists of two codes and two carrier phase observations in RINEX format which
were considered. This observations can be formulated as follows (e.g. Leandro, 2009; e.g. Sedeek et al., 2017):
$P1 = R + c.(dT - dt) + T + I + HDr,1 - HDs,1 + M1 + E1$             *(1)*
$P2 = R + c(dT - dt) + T + \gamma I + HDr,2 - HDs,2 + M2 + E2$             *(2)*
$\Phi1 = R + c(dT - dt) + T - I + \lambda1N1 + pbr,1 - pbs,1 + hdr,1 - hds,1 + m1 + e1$             *(3)*
$\Phi2 = R + c(dT - dt) + T - \gamma I + \lambda2N2 + pbr,2 - pbs,2 + hdr,2 - hds,2 + m2 + e2$             *(4)*
Where:

    *P1* and *P2*      Pseudo-range measurements on L1 and

                      L2 frequencies, respectively, in meter;

    *Φ1* and *Φ2*      carrier-phase measurements on L1 and L2 frequencies, respectively, in meter;

    *R*              the geometric distance between satellite and receiver antennas, in meters



| $C$ | the speed of light, in meters per second; |
|---|---|
| $dT$ and $dt$ | receiver and satellite clock errors, respectively, in seconds; |
| $T$ | the neutral troposphere delay, in meters; |
| $I$ | the L1 frequency ionosphere delay, in meters; |
| $\Gamma$ | the factor to convert the ionospheric delay from L1 to L2 frequency, |
| $N1$ and $N2$ | carrier-phase integer ambiguities on L1 and L2 frequencies, respectively, in cycles; |
| $\lambda1$ and $\lambda2$ | carrier-phase wave length on L1 and L2 frequencies, respectively, in meters; |
| $HDr,i$ and $HDs,i$ | receiver and satellite pseudo-range hardware delays, respectively, in metric units, where i represents the frequency (L1 or L2); |
| $hdr,i$ and $hds,i$ | receiver and satellite carrier-phase hardware delays, respectively, in metric units, where i represents the frequency (L1 or L2); |
| $M1$ and $M2$ | Pseudo-range multipath on L1 and L2 frequencies, respectively, in meters; |
| $E1$ and $E2$ | Other un-modeled errors of pseudo-range measurements on L1 and L2 frequencies, respectively, in meters. |
| $pbr,i$ and $pbs,i$ | receiver and satellite carrier-phase initial phase bias, respectively, in metric units, where i represents the carrier frequency (L1 or L2); |
| $m1$ and $m2$ | carrier-phase multipath on L1 and L2 frequencies, respectively, in meters; |
| $e1$ and $e2$ | Other un-modeled errors of carrier-phase measurements on L1 and L2 frequencies, respectively, in meters. |


## 3. Fixing Ambiguity using Sequential Least Square

As it is shown in the previous section, estimating the ionospheric delay in its most accurate values, by using GPS
carrier-phase measurements, depends mainly on the ability to determine the ambiguity term in equation (3, 4). To fix the
ambiguities to its most accurate float values, a combination algorithm of Ionosphere-free and Geometry-free was used as
follows (Guochang Xu, 2004):
$$\begin{bmatrix} \lambda_1 N_1 \\ \lambda_2 N_2 \\ B1 \\ \varepsilon \end{bmatrix} = \begin{bmatrix} 1-2a & -2b & 0 & 2 \\ -2a & -2a-1 & 0 & 1 \\ 1/q & -1/q & 0 & 0 \\ a & b & 0 & 0 \end{bmatrix} \begin{bmatrix} P1 \\ P2 \\ \Phi1 \\ \Phi2 \end{bmatrix} \tag{5}$$

**Where:**
$a = f_1^2/(f_1^2 - f_2^2),$ $\qquad b = -f_2^2/(f_1^2 - f_2^2),$
$q = f_1^2 * (1/f_1^2 - 1/f_2^2).$
$\varepsilon = R + c.(dT - dt) + T + m_j + e_j, j=p;\Phi,$
$I = B1,$ $B1$ is ionospheric parameters in the path and zenith,
$f_i^2$:Frequency where i: 1; 2.



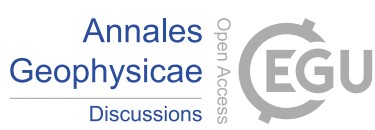

The ionospheric delay $B1$ which would be a check of our autonomous ionospheric delay $I_{1(t_0,t_{0+n})}$ Due to the fixed values of
the ambiguity term of each satellite as long as no slips are occurred, a Sequential Least Square Adjustment can be applied for
each additional observation. This is done by partitioning of the observation model as represented in following forms
(Guochang Xu, 2004):
$L=AX$ and $L = \begin{bmatrix} L1 \\ L2 \end{bmatrix}$ , $n = \begin{bmatrix} n1 \\ n2 \end{bmatrix}$, $A = \begin{bmatrix} A1 \\ A2 \end{bmatrix}$ and $P = \begin{bmatrix} P1 \\ P2 \end{bmatrix}$     (6)
Where: $L$ is the vector of observations of the ambiguity term, $A$ is the design matrix, $X$ is unknown parameters vector, $n$ is
the vector of unknowns and $P$ is weight matrix of observations with size $n*n$.
$L_1$ and $L_2$ are first and second groups of observations which its number $n_1$ and $n_2$. The design matrix $A_1$ and $A_2$ and its weight
matrix $P_1$ with dimension $n_1*n_1$ and $P_2$ with dimension $n_2*n_2$ respectively. We suppose that $n_1$ is the first five epochs and a
preliminary solution $x_o$ can be calculated as follows:
$x_o = (A_1^T P_1 . A_1)^{-1} A_1^T . P_1 . L_1$          (7)
The change due to the additional observation set $L_2$ is denoted as $\Delta x$:
$\Delta x = (A_1^T P_1 . A_1 + A_2^T P_2 . A_2) - 1. A_2^T P_2 (L_2 - A_2 . x_o)$   (8)
$X = x_o + \Delta x$          (9)
By using this system of equations, final float ambiguities values
are computed.

## 4. Ionospheric Delay Estimation by Geometry-Free Linear Combination of GPS
Observables
The geometry-free linear combination of GPS observations is classically used for ionospheric investigations. It can be
obtained by subtracting simultaneous pseudo range (P1-P2 or C1-P2) or carrier phase observations (Φ1-Φ2). With this
combination, the satellite – receiver geometrical range and all frequency independent biases are removed (Ciraolo et al.,
2007). Subtracting Eq. (3) from Eq. (4), the geometry-free linear combination for carrier phase observations is obtained (e.g.
Sedeek et al, 2017):
$= (\gamma - 1) I_1 + (\lambda_1 N_1 - \lambda_2 N_2) + c(\Delta hdr - \Delta hds) + \varepsilon_{(\Phi_1 - \Phi_2)}$   (10) $L_{4(t)} = \phi_{GF} = \phi_{L1} - \phi_{L2}$
Where:
$L_4$ at epoch (t) is the carrier phase Geometry free ($\Phi_{GF}$)
$\varepsilon_{(\Phi_1 - \Phi_2)}$ is the noise term in phase equation can be neglected for the sake of simplicity,
$\Delta hdr$ is the difference carrier-phase hardware delays bias between $L1 \& L2$ frequency for receiver.
$\Delta hds$ is the difference carrier-phase hardware delays bias between $L1 \& L2$ frequency for satellite.
the factor γ is the factor to convert the ionospheric delay from $L1$ to $L2$ frequency
$I_2 = \dfrac{40.3 \; STEC}{f_2^2} = \dfrac{f_1^2}{f_2^2} I_1 = \gamma I_1 , \gamma = \dfrac{f_1^2}{f_2^2}$          (11)
$DCBs = C \times (\Delta hdr - \Delta hds)$     (12)
Differential Code Bias (*DCBs)* of satellites and receivers are the main source of errors in estimating total electron content
(*TEC*) using GPS measurements. Ignoring DCBs of satellites and receivers can cause an error in estimating *TEC* reaches up
to several nanoseconds (ns) (Sardon and Zarraoa 1997).
In the current paper, DCBs of satellites and receiver were considered from IGS products (ION files), and Melbourne-Wübbena
(1985) Linear Combination is utilized to detect and repair cycle slips.
The ionospheric residual between epoch $t_o \& t_{o+k}$ can be computed by subtracting the carrier phase Geometry free ($\Phi_{GF}$) at
both epochs (equation 10) is sufficient for estimating ionospheric delay then *VTEC* estimation (Araujo-Pradere et al., 2007).
$\partial L_{4(t_0,t_{0+k})} = L_{4(t_{0+k})} - L_{4(t_0)} = \dfrac{f_1^2 - f_2^2}{f_1^2} \partial I_{1(t_0,t_{0+k})}$          (13)

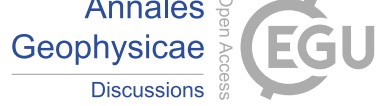



Thus, $I_{1(t_{0+n})} = I_{1(t_0)} + \sum \partial I_{1(t_0,t_{0+n})}$      (14)
Where: $k$ is epoch number ($t_0 < k < n$) and $I_{1(t_0)}$ is the ionosphere delay at the first epoch ($t_o$). By substituting equation 13 & 14
in equation 10, one can get the observation equation at epoch ($t_o + n$) as:
$L_{4(t_{0+n})} - \frac{f_1^2 - f_2^2}{f_1^2} \sum \partial I_{1(t_0,t_{0+n})} = (\lambda_1 N_1 - \lambda_2 N_2) + \frac{f_1^2 - f_2^2}{f_1^2}\left(I_{1(t_0)}\right) + DCBs$    (15)

The epoch-wise carrier phase Geometry free $L_{4(t_{0+n})}$ at eq. (15) includes two parts, one is due to the spatial and temporal
change of the ionospheric delay from epoch ($t_o$) to ($t_o + n$), and second part is related to the ambiguity term ($\lambda_1 N_1 - \lambda_2 N_2$)

**5. Elevation and Azimuth Angles and Ionosphere Pierce Point:**
satellite elevation and azimuth angle can be computed by converting receiver position from Earth Centered Earth
Fixed (ECEF) to geodetic coordinate $(\lambda, \varphi, z)$. Then, interpolating satellite position coordinate $(x_s, y_s, z_s)$ at the
specified epoch from the IGS final orbits. The interpolated satellite position is then transformed to a local coordinate frame,
East, North, and Up (ENU) system. The transferred ENU is used to calculate elevation and azimuth angles as follows (e.g.
Dahiraj, 2013):
$E = \arctan\left(\frac{x_U}{\sqrt{x_N^2 + x_E^2}}\right)$      (16)
$A = \arctan\left(\frac{x_E}{x_N}\right)$      (17)
Where: $E$ and $A$ is elevation angle and Azimuth angle of satellite, respectively.
Ionospheric Pierce Point (IPP) can be estimated by using reference station coordinates $(\phi_r, \lambda_r)$, so the geographic latitude
and longitude of IPP can be computed according to elevation and azimuth angle of satellite such as follows (e.g. Dahiraj,
150    2013):

$\psi = \pi / 2 - E' - E$      (18)
Where:
$\psi$ : The offset between the IPP and the receiver;
$E'$ and $E$: the elevation angles at the IPP and receiver.
$E' = \sin^{-1}\left(\left(\frac{R_E}{R_E + H}\right)\cos E\right)$      (19)
$R_E$: is the mean radius of the spherical Earth (6371 km)
$H$: is the height of IPP (it is taken to be 450 km)
the ionosphere assumed to be concentrated on a spherical shell of a thin layer located at altitude of 450 km above Earth's
surface, i.e., forming single layer model (Rocken et al., 2000). IPP is the intersection point between the satellite receiver line-
of-sight, and the ionosphere shell see Figure (1). Slant total electron content (*STEC*) can be translated into *VTEC* using Single
Layer Model (SLM)
















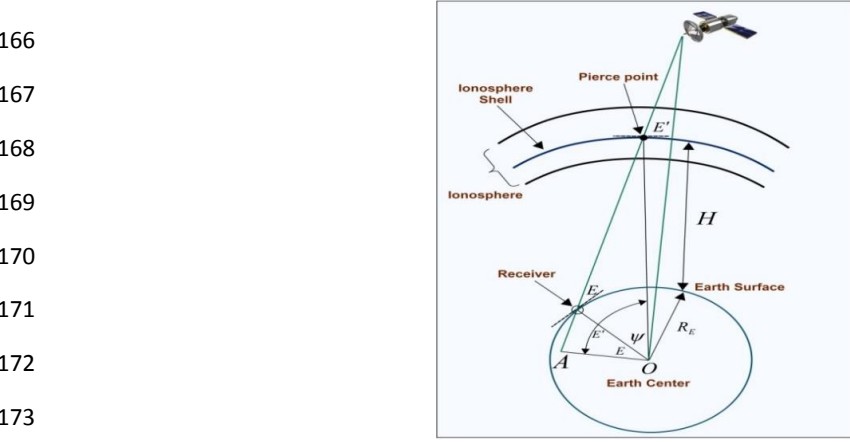

174         Figure (1): Elements of the spherical ionospheric shell model (Sedeek et al.,2017).


$$VTEC = F(E)\,STEC \qquad (20)$$
$$F(E)=\frac{1}{\cos(E')} \qquad (21)$$
From equations (5 & 20)
$$VTEC = F(E)\frac{I_1 \cdot f_1^2}{40.3} \qquad (22)$$
## 6. Evaluation of the Study
To evaluate the performance of the developed algorithm, VTEC maps cover a 24 hours at intervals of 2 hours were estimated
for observations of Day 98, 2015 for two IGS stations ANKR, and BSHM as shown in figure (2). A geographic reference
frame was used to produce the epoch-specific instantaneous regional maps of the ionosphere. Figures (3) and (4) show the
regional ionosphere maps for both stations using the proposed code.
It should be noted that the IGS GIMs are computed from a Global IGS network, with observation of a combination of ~ 500
permanent GNSS stations. It is observing 4~12 satellites at 30 sec rates. This means that more than 20,000 VTEC
observations/hour worldwide. It uses Kalman-filter approach with shell model ionosphere with slab centered 450 km. It
produces VTEC maps for every 2 hours from $180^{\circ}$W to $180^{\circ}$ E with $5^{\circ}$ spatial resolution in longitude and from $87.5^{\circ}$S to
$87.5^{\circ}$N with $2.5^{\circ}$ spatial resolution in latitude (Krankowski, 2016). The IGS GIMs are provided by several analysis centers
(ACs). We select a region located between $20^{\circ}$~ $45^{\circ}$north geographic latitude and $20^{\circ}$~ $45^{\circ}$ longitude. This region covers the
IPPs location for most of the processed epochs.












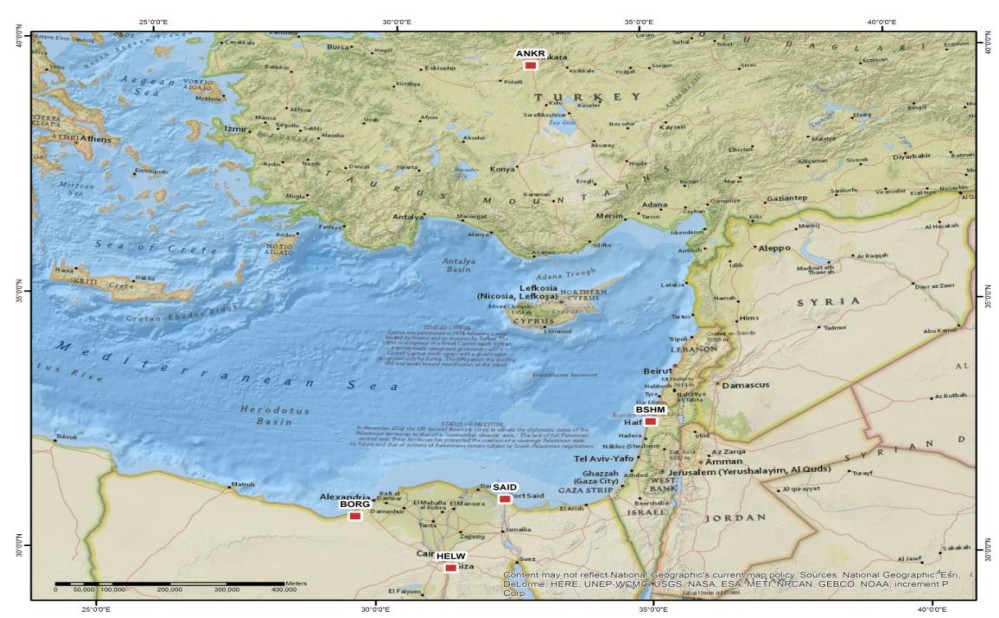

212        Figure (2) Location of the three Delta Stations, Egypt, and IGS stations used to estimate the regional *TEC* values.









Figure (3) TEC maps over ANKR
station of Day 98 (2015)








241          Figure (4) TEC maps over BSHM station of Day 98 (2015)







Geographic latitude

Geographic longitude

244        Figure (5). TEC maps estimated by GIM model of day98 (2015)





## 7. Results and Discussions

The observations of Day 98, 2015 for ANKR, BSHM IGS stations were used to evaluate the performance of the proposed algorithm. Figures (3) and (4) present the regional ionosphere maps cover a 24 hourly period at intervals of 2 hours for both stations using the ZDPID code with cut off angle 10˚. the selected region located between 20˚~ 45˚N and 20˚~ 45˚ E. As it mentioned before, this region covers the IPPs location over the two stations. Figure (5) shows GIM for day 98, 2015. The obtained mean TEC values every 2 hours of the two IGS stations using the ZDPID are shown in figure (6). The difference between the estimated VTEC over ANKR station and GIM is ranges from -2.1 to 3.67 TECU and ranges from -0.29 to 3.65 TECU between estimated VTEC over BSHM station and GIM.IGS Analysis centers (ACs) often use TEC representation algorithms, which result in a model resolution comparable with the whole area of the region under investigation (Schaer, 1999). Wielgosz et al., (2003b) presented an example ionosphere maps for the Ohio CORS compared to the global GIMs. The GIMs general TEC level is higher by about 3-5 TECU, as compared to the maps generated using the Kriging and Multiquadric methods.

Regional ionosphere maps have been generated for both quiet and stormy days by using Bernese 5.0 PPP model by Salih Alcay et al. (2012). They found that the biggest difference between single stations based regional model and GIMs are about 6 TEC in quiet day. The GIMs suffered from the lack of stations at some areas e.g., over the oceans, e.g. lack of data over the equatorial, North Africa, Atlantic and in-part over equatorial and southern Pacific. This shortage of data, hamper the detection of the equatorial anomalies (Krankowski, 2016). To overcome this shortage over Egypt and similar territories, the developed algorithm in this paper is applied to improve the temporal and spatial resolution of GIM. Figure (2) shows three GPS receivers (BORG, HELW and SAID) were considered to estimate VTEC over Nile delta .

In the current study, the satellite DCB for the three stations was derived from IGS products (IONEX files) and the receiver DCBs were calculated by (Sedeek et al., 2017). Regional VTEC maps are drawn on a ($0.5° \times 0.5°$ grid) (Figures 7, 8, and 9). Figure (10) illustrate the mean values of VTEC every 2 hours for the three GPS stations and GIMs. As it is seen in the figure, the mean VTEC value of GIM is higher than the three stations at the most time. The difference ranges from - 1.1 to 3.69 TECU between GIM and SAID station, from -1.29 to 3.27 TECU between GIM and HELW station, and from 0.2 to 4.2 TECU between GIM and BORG station.

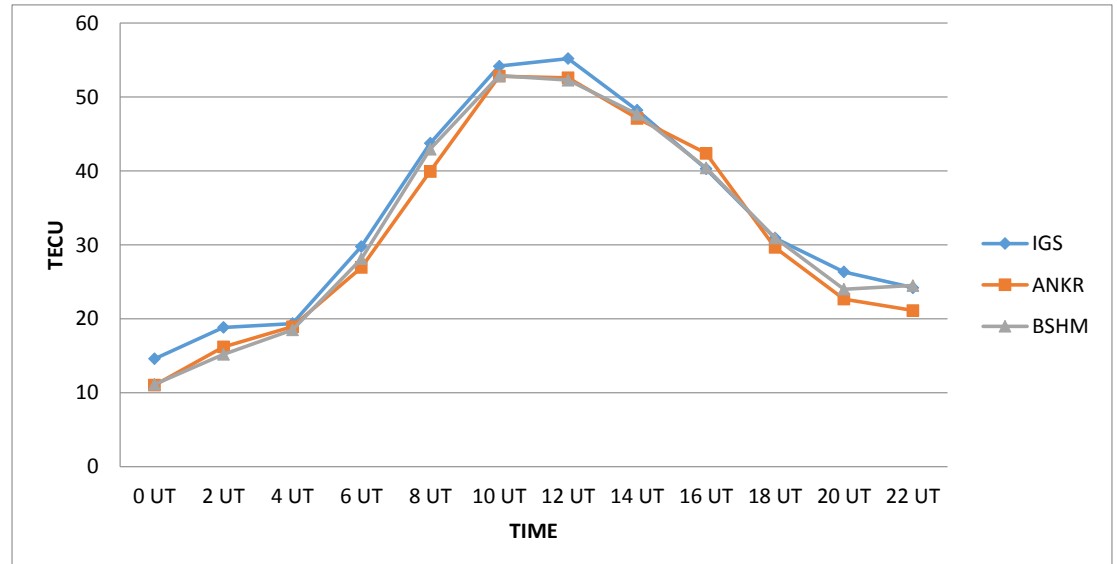

Figure (6). Mean TEC results every 2 hours of the IGS station and IGS GIMs





Figure (7). TEC maps over HELW station, Day 98-2015








Figure (8). TEC maps over BORG station, Day 98-2015





Figure (9). TEC maps over SAID station, Day 98-2015.






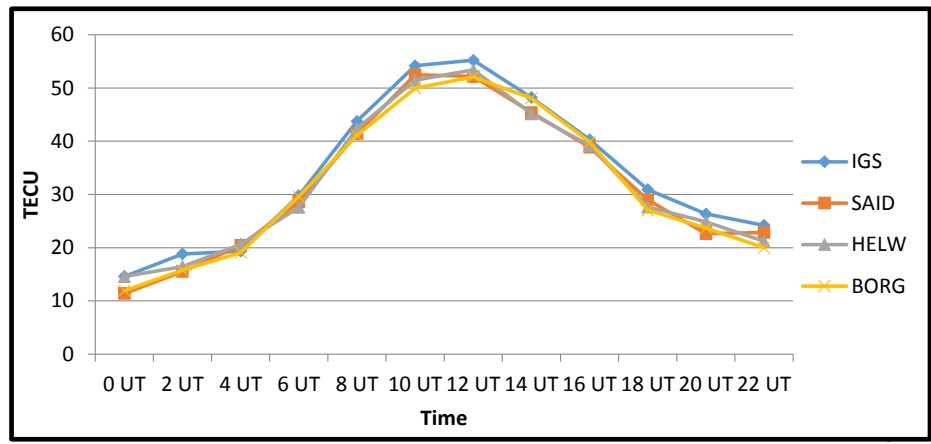

Figure (10). Mean TEC results every 2 hours of the Delta stations and IGS GIMs

## 8. Conclusions

*TEC* maps are needed to characterize the ionospheric behavior for satellite based positioning. Many studies have been performed for *TEC* mapping. In the present study an algorithm was developed to estimate regional VTEC maps using a single station (Zero-difference) based on carrier phase observations. The core of this algorithm is mainly depending on computing ambiguity term in carrier phase observations using Sequential Least Square Adjustment. The proposed algorithm has been written using MATLAB code. GIMs suffers from deficiency of GPS stations at some areas (e.g., over the oceans), e.g. lack of data over the equatorial, North Africa, Atlantic and in-part over equatorial and southern Pacific. This shortage of data, hamper the detection of the equatorial anomalies. To overcome this deficiency, improving the temporal and spatial resolution of GIM, and monitoring VTEC over Egypt, the developed algorithm in this study has been applied.

To evaluate the developed algorithm, the V*TEC* values have been estimated over two stations from the IGS network, namely ANKR and BSHM. *GIMs* were used as a threshold values for comparison with the estimated VTEC from the proposed algorithm. the mean V*TEC* value of *GIM* is higher than the two stations at most time. The difference ranges from 0.46 to 3.8 TECU between *GIM* and ANKR station. and from 0.1 to 3.65 TECU between BSHM station and GIM, The difference ranges from - 1.1 to 3.69 TECU between *GIM* and SAID station, from -1.29 to 3.27 TECU between GIM and HELW station, and from 0.2 to 4.2 TECU between GIM and BORG station.

ZDPID gives a much more detailed picture and real perception of the local ionosphere map from single point. On the other hand, results of the verification process are clearly shown that the developed algorithm can be successfully used for generating regional ionosphere maps.

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
