# Peer review of "Regional Ionosphere Mapping Using Zero Difference GPS 1 Carrier Phase 2 Heba Tawfeek1, Ahmed Sedeek2, Mostafa Rabah3, Gamal El-Fiky1 3 4 1. Faculty of Engineering, Zagazig University, Egypt, 5 2. Higher Institute of Engineering and Technology, El-B"

_Annales Geophysicae, 2018_

## Referee Comment (RC1) · Anonymous Referee #1 · 8 Jan 2019

ANGEO-2018-121 Title: Regional Ionosphere Mapping Using Zero Difference GPS Carrier Phase Authors: Heba Tawfeek et al.

General comments: This manuscript presented an algorithm capable of producing local ionosphere map using carrier phase observations over a single GPS station. In my opinion, this manuscript doesn't contain new ideas, and the experiment is too simple and can't support your conclusions. 1. The algorithm is new? What is the detail information about the ZDPID software? 2. There is no information about your local ionosphere model, spherical harmonic expansion? two-dimensional Taylor series expansion? 3. You still use the code observations in your algorithm. 4. You selected two IGS stations to model the local ionosphere map, why you selected another three stations in the same way? 5. The GIM is generated with about 300 world wide GNSS

stations, why you compare the GIM with your local ionosphere map? Yes, there will be differences in this comparison, but how do you convince me that your result is correct? The assessment method is not appropriate. 6. Figure 3-4, 7-9 can't represent the local ionosphere VTEC well. Figure 6 and 10 should be plotted with professional software. 7. How does your algorithm in your paper improve the temporal and spatial resolution of GIM.

Specific comments: 1. The title "Regional ionosphere mapping using zero difference GPS carrier phase" may be not appropriate. "Regional" can be replaced by "local", "zero difference" can be replaced by "a single station". 2. In the Abstract part, P1. Line 25-38 should be rewritten. 3. In the Introduction part, there is no information about the ionosphere mapping with carrier phase observations. 4. P.2 Line 42, 45, introduce all the abbreviations, e.g. GNSS, VTEC. . . .. . . 5. P.2 Line 73, what is the symbol gama? 6. Equation (5), 1-2a, -2b, 0, 2 in your manuscript is different from 1-2a, -2b, 1, 0 in Xu, 2004, what is the difference? 7. P.3 Line 81, What do you mean "To fix the ambiguities to its most accurate float values"? In general, we fix the ambiguities means that the ambiguities are set as integer values. 8. P.4. Line 95, what is the difference between X and n? 9. P.4 Line 114, equation (10) ? 10. Equation (12) DCB is differential code bias, not the differential carrier phase hardware delay bias. 11. P.4 Line 126. DCBs from IONEX file is for code observations not for carrier phase. 12. Section 5 is to general, can be deleted. 13. P.11 Line 250, what do you mean "the mean TEC values"? above the station? Over the local area?

Technical corrections: 1. This manuscript is not presented with a professional format. It is very hard for me to read and to understand your meaning. 2. P.11 Line 251 "is ranges from . . . . . ."

---

## Referee Comment (RC2) · Anonymous Referee #2 · 9 Mar 2019

The manuscript reports development of an algorithm capable of producing hourly VTEC regional maps using carrier phase observations from a single station, dual frequency GPS receiver. In order to achieve this objective, the authors have used Sequential Least Square Adjustment (SLSA) method to fix the carrier phase ambiguity and developed a MATLAB code named ZDPID. Data from two IGS stations (ANKR and BSHM) are used to develop this code and the outputs are compared with the Global Ionospheric Maps (GIM) to gain confidence. Afterwards, GPS observations from three stations in over Nile Delta in Egypt (SAID, HELW and BORG) are used to generate the regional ionospheric maps for three cases and they found the comparison of these outputs with the IGS VTEC satisfactory. The importance of the work lies in its supposed ability to reproduce regional TEC maps using single station data. I have several

concerns regarding this work and would like the authors to be more critical in claiming the targeted objective. The concerns are as follows.

1. GIMS are uncertain themselves! What are typical uncertainties in GIMs? When the authors use comparison with the GIMs to validate their code, they are already comparing with gross maps! Therefore, they need to be careful with the mean difference values. 2. ZDPID code uses cut-off angle of 10 deg. Therefore, the data that go into the ZDPID contain huge uncertainty due to multi-path errors etc. 3. The final outputs for the three stations are significantly different amongst themselves on many occasions. Yet, they match very well with the IGS values! How? For what latitude/longitude bin? 4. The authors should think of using large scale ionospheric features like plasma fountain over the African sector to validate the maps from these three stations. At present the work does not use any physical process to validate the TEC maps. 5. What is rational of taking IPP at 450 km and not at 350 km? 6. The abstract is lengthy, contain general information. There is no discussion of earlier works and the novelty of the present work. Is similar approach not followed earlier? 7. Put error bars in Figures 6 and 10. Mention which lat/long bin are you comparing? 8. Figure 2 is not readable.

I am not able to recommend this work for publication in the present form.